# Therapy-Induced Stromal Senescence Promoting Aggressiveness of Prostate and Ovarian Cancer

**DOI:** 10.3390/cells11244026

**Published:** 2022-12-13

**Authors:** Elisa Pardella, Erica Pranzini, Ilaria Nesi, Matteo Parri, Pietro Spatafora, Eugenio Torre, Angela Muccilli, Francesca Castiglione, Massimiliano Fambrini, Flavia Sorbi, Paolo Cirri, Anna Caselli, Martin Puhr, Helmut Klocker, Sergio Serni, Giovanni Raugei, Francesca Magherini, Maria Letizia Taddei

**Affiliations:** 1Department of Experimental and Clinical Biomedical Sciences “Mario Serio”, Viale Morgagni 50, 50134 Florence, Italy; 2Department of Minimally Invasive and Robotic Urologic Surgery and Kidney Transplantation, University of Florence, 50134 Florence, Italy; 3Department of Health Sciences, Section of Pathology, University of Florence, 50134 Florence, Italy; 4Histopathology and Molecular Diagnostics, Careggi Teaching Hospital, 50134 Florence, Italy; 5Department of Urology, Division of Experimental Urology, Medical University of Innsbruck, A-6020 Innsbruck, Austria; 6Department of Experimental and Clinical Medicine, University of Florence, Viale Morgagni 50, 50134 Florence, Italy

**Keywords:** prostate cancer, ovarian cancer, therapy induced senescence, tumor microenvironment, docetaxel, cisplatin

## Abstract

Cancer progression is supported by the cross-talk between tumor cells and the surrounding stroma. In this context, senescent cells in the tumor microenvironment contribute to the development of a pro-inflammatory milieu and the acquisition of aggressive traits by cancer cells. Anticancer treatments induce cellular senescence (therapy-induced senescence, TIS) in both tumor and non-cancerous cells, contributing to many detrimental side effects of therapies. Thus, we focused on the effects of chemotherapy on the stromal compartment of prostate and ovarian cancer. We demonstrated that anticancer chemotherapeutics, regardless of their specific mechanism of action, promote a senescent phenotype in stromal fibroblasts, resulting in metabolic alterations and secretion of paracrine factors, sustaining the invasive and clonogenic potential of both prostate and ovarian cancer cells. The clearance of senescent stromal cells, through senolytic drug treatment, reverts the malignant phenotype of tumor cells. The clinical relevance of TIS was validated in ovarian and prostate cancer patients, highlighting increased accumulation of lipofuscin aggregates, a marker of the senescent phenotype, in the stromal compartment of tissues from chemotherapy-treated patients. These data provide new insights into the potential efficacy of combining traditional anticancer strategies with innovative senotherapy to potentiate anticancer treatments and overcome the adverse effects of chemotherapy.

## 1. Introduction

Cellular senescence is a state of stable and irreversible cell cycle arrest induced by telomere shortening or stressful insults, such as oncogene activation, DNA damage, epigenetic perturbations, or oxidative stress [1,2,3]. Over the last decades, cellular senescence has been related to a broad spectrum of age-related disorders, including atherosclerosis, heart failure, pulmonary insufficiency, and neurodegeneration [4,5,6]. In the context of cancer, senescence represents a double-edged sword [7]: on one side, it plays a tumor-suppressive role by preventing the proliferation of cells during the pre-malignant transformation steps [8,9]; on the other side, the chronic accumulation of senescent cells can sustain tumor progression, metastasis, immunosuppression and cancer relapse [10,11]. In particular, the senescence-associated secretory phenotype (SASP), a collection of growth factors, cytokines, and proteases secreted by senescent cells, exerts both autocrine and paracrine effects, thus profoundly remodeling the tumor microenvironment (TME) [12].

Indeed, the tumor stroma plays a pivotal role in cancer progression by establishing reciprocal and dynamic crosstalk with cancer cells promoting the acquisition of cancer malignancy [13], including the development of therapeutic resistance [14,15]. Within the tumor stroma, cancer-associated fibroblasts (CAFs) represent a leading cell type responsible for a large variety of pro-tumorigenic signals [16,17,18]. Notably, several studies reported that senescent stroma supports different steps of cancer progression [19,20].

In particular, senescent fibroblasts have been demonstrated to elicit the proliferation of pre-neoplastic prostate epithelial cells in vitro and promote tumorigenesis in in vivo models [21]. Accordingly, replicative- and H_2_O_2_-induced senescent fibroblasts sustain the acquisition of aggressive traits in prostate cancer [22], and trigger a neoplastic transformation of dormant, partially transformed ovarian epithelial cells [23].

Noteworthy, several anticancer interventions, including common chemotherapeutic drugs, radiotherapy, and immunotherapy, may induce senescence in both cancer cells and cellular components of the TME, a phenotype termed “therapy-induced senescence” (TIS) [24,25,26,27]. On one side, TIS may represent a mechanism through which genotoxic agents promote cancer regression by inducing the senescent phenotype in cancer cells [28]. However, TIS also contributes, through the release of SASP factors, to many of the detrimental side effects of anticancer therapies, including cardiac dysfunction, fatigue, bone loss, and tumor relapse [29,30]. Despite the large body of evidence showing TIS effects on tumor cells, little is known about the induction of the senescent phenotype triggered by anticancer interventions in the stromal compartment [31,32]. A recent study highlighted that upon irradiation, CAFs are predisposed to p53-mediated TIS, that in turn leads to tissue remodeling and chemoradiotherapy resistance in a murine rectal cancer model [33]. Besides radiotherapy, different classes of chemotherapeutic agents promote the senescent phenotype in non-cancerous cells [29,34,35,36,37], resulting in increased tumor growth, metastasis formation, cancer relapse [38], and chemoresistance [39].

The taxane drug Docetaxel (DTX) and the alkylating agent Cisplatin (CPT) are chemotherapeutics that respectively act by inhibiting the proper assembly of microtubules into the mitotic spindle and by interfering with DNA replication [40,41]. These compounds are standard cancer therapy for several tumor types, including metastatic castrate-resistant prostate cancer (DTX) [42] and ovarian cancer (CPT) [43]. Despite their wide applications, DTX and CPT-treated patients often experience chemotherapy resistance [44,45]. Therefore, identifying the adverse bystander effects correlated to DTX or CPT treatment represents an unmet challenge to develop novel strategies targeting chemotherapy vulnerabilities.

Here, we found that treatments with DTX and CPT strongly promote the acquisition of the senescent phenotype in in vitro prostate and ovarian patient-derived fibroblasts. Remarkably, lipofuscin staining of paraffin-embedded tissue sections from post-treatment patients with prostate or ovarian cancer revealed an increase of senescence in the stromal compartment after DTX or CPT exposure. Moreover, we observed these senescent fibroblasts, by rewiring their metabolism and activating the SASP, enhance the malignant behavior of prostate and ovarian cancer cells. Significantly, this detrimental crosstalk between tumor cells and senescent fibroblasts can be reverted by clearance of senescent fibroblasts through treatment with a senolytic drug.

Concluding, our data further enforce the recently proposed “one-two punch strategy”, which consists of the sequential treatment with a classic anticancer intervention followed by senescence-targeting compounds [46,47,48,49]. In this context, our results pave the way for the development of novel therapeutic approaches based on the eradication of the senescent stromal components to overcome the detrimental bystander effects of chemotherapy, thus improving its efficacy in cancer patient treatment.

## 2. Materials and Methods

### 2.1. Cell Lines

Human prostate (PC3 and DU-145) and ovarian (A2780 and SKOV-3) cancer cell lines were obtained from ATCC. Human Prostate Fibroblasts (HPFs) were isolated from surgical explants of patients (average age 70) subjected to surgical intervention for lower urinary tract symptoms due to benign prostatic hyperplasia (BPH). Final histology confirmed BPH for all samples. Human Ovarian Fibroblasts (HOFs) were isolated from surgical explants of healthy peritoneal tissues from patients (average age 66) subjected to surgical intervention during cytoreductive surgery for advanced ovarian cancer. Surgical explants were obtained in accordance with the Ethics Committee of the Azienda Ospedaliera Universitaria Careggi (Florence, Italy).

All cells were cultured in Dulbecco’s Modified Eagle Medium (DMEM) high glucose (4.5 g/L) (Merck Sigma, St. Louis, MO, USA, #D5671), supplemented with 10% Fetal Bovine Serum (EuroClone, Milan, Italy #ECS0180L), 2 mM L-glutamine (Merck Sigma #G7513), 1% penicillin and streptomycin (EuroClone #ECB3001D) (complete medium). Cells were routinely grown at 37 °C in a humidified atmosphere with 5% CO_2_. Specifically, HPFs and HOFs were isolated as follows. A fragment of surgical explant conserved in PBS was transferred into a cell culture plate and cut into small pieces of about 0.2 cm with a scalpel blade. The obtained pieces were transferred into a new cell culture plate and placed at a distance of about 0.5–1 cm in a central stripe (8–10 pieces in each cell culture plate). A sterilized slide was put above the pieces and pressed on them. Then, DMEM High Glucose, supplemented with 20% FBS, 2 mM L-glutamine, 2% penicillin/streptomycin, 100 μg/mL Kanamycin (Merck Sigma), and 2.5 μg/mL Amphotericin B (Merck Sigma), was added to the fragments. After 20–30 days, fibroblasts, that had formed a monolayer, were detached by trypsinization and routinely cultured as previously described.

### 2.2. Cell Treatments and Preparation of Conditioned Media

To induce senescence, HPFs and HOFs were treated for 24 h with DTX 5 nM in DMSO (Merck Sigma #01185) or CPT 20 μM in DMSO (Merck Sigma #5663-27-1), respectively, while DMSO (Merck Sigma #472301) was added in control samples. Cells were then incubated with fresh culture media for additional 6 days, before performing experiments, unless otherwise specified. For treatment with senolytic, HPFs and HOFs were incubated for 24 h with DTX or CPT, then for 1 day with fresh culture medium, and finally for 72 h with 1.25 μM ABT263 (Navitoclax) in DMSO (Selleckchem.com #S100), before performing experiments. To collect conditioned media (CM), senescent cells obtained as previously described were incubated for 24 h with serum-free media. After that, CM was collected, clarified by centrifugation for 10 min at 1000 rpm, and used fresh or stored at −80 °C until use. Prostate and ovarian cancer cells were incubated for 72 h with CM from HPFs or HOFs, respectively.

### 2.3. Proliferation Assay

A 3 × 10^5^ HPFs/well and 3 × 10^5^ HOFs/well were seeded in 35-mm culture dishes. After 24 h, corresponding to the “time zero” of each experiment, cells were treated with DTX 5 nM or CPT 20 μM for 24 h. The culture media was then replaced with fresh complete DMEM. At each time point, cells were harvested by trypsinization (Trypsin EDTA, EuroClone #ECM0920D), resuspended in a complete medium, and counted.

### 2.4. Senescence-Associated β-Galactosidase Staining

HPFs and HOFs were fixed with 3% paraformaldehyde in PBS for 5 min and then incubated at 37 °C in a non-humidified incubator under atmospheric CO_2_ conditions for 16 h in freshly prepared senescence-associated β-galactosidase (SA-β-Gal) staining solution (5 mM potassium ferrocyanide, 5 mM potassium ferricyanide, 150 mM NaCl, 2 mM MgCl_2_, 40 mM citric acid monohydrate, 1 mg/mL 5-bromo-4-chloro-3-indolyl β-D-galactopyranoside (X-Gal; Merck Sigma #B4252), pH 6.0). Photos at ten randomly chosen fields for each sample were taken (Nikon Eclipse TS100 inverted microscope) and cells positive to SA-β-Gal staining were detected by the presence of an insoluble blue intracellular precipitate. Total and positive cells were counted using the ImageJ imaging system.

### 2.5. Immunocytochemistry

Glass coverslip-plated HPFs were fixed with 3% paraformaldehyde in PBS for 20 min at 4 °C. After several washes with PBS, cells were permeabilized with TBST (50 mM Tris-HCl pH 7.4, 150 mM NaCl, 0.1% Triton X-100) and then incubated for 1 h with 5.5% horse serum in TBST. Cells were incubated overnight at 4 °C with the anti-phospho-histone H2AX (S139) antibody (Abcam, Cambridge, UK, #81299), diluted 1:100 in TBS (50 mM Tris-HCl pH 7.4, 150 mM NaCl) containing 3% bovine serum albumin (BSA; Merck Sigma #A7906). After washing for 15 min with TBST and for an additional 15 min with TBST containing 0.1% BSA, HPFs were incubated for 1 h with the fluorescent anti-rabbit Alexa Fluor 488 secondary antibody (Life Technologies Invitrogen, Waltham, MA, USA, #A-11034), diluted 1:500 in TBS containing 3% BSA. Nuclei were stained by incubating cells with 200 nM DAPI (Merck Sigma #D9542). Cells were examined using TCS SP8 confocal microscope (Leica, Wetzlar, Germany).

### 2.6. Western Blotting

Cells were lysed in RIPA lysis buffer (50 mM TrisHCl pH 7.5, 150 mM NaCl, 100 mM NaF, 2 mM EGTA, 1% triton X-100), supplemented with protease (Merck Sigma #P8340) and phosphatase (Merck Sigma #P5726) cocktail inhibitors. Protein lysates were centrifuged at 14,000 rpm at 4 °C for 10 min and protein concentration was quantified using the bicinchoninic acid (BCA) assay (Merck Sigma #BCA1). 20 to 50 μg of total proteins were loaded on SDS-PAGE gels and transferred to polyvinylidene difluoride (PVDF) membranes (BioRad, Hercules, CA, USA, #1704157). Membranes were activated with methanol and incubated for 1 h in a blocking buffer (5% non-fat dry milk (SantaCruz Biotechnology, Dallas, TX, USA, #sc-2325) in PBS-Tween 0.1%). Then, membranes were incubated overnight at 4 °C with the appropriate primary antibodies, diluted 1:1000 in PBS-Tween 0.1% containing 5% BSA. The primary antibodies were from Santa Cruz Biotechnology: anti-p16 (# sc-56330); anti-p21 (# sc-271610); anti-actin (# sc-47778) antibodies.

The following day, after washing in PBS-Tween 0.1%, membranes were incubated with horseradish peroxidase-conjugated anti-mouse (SantaCruz Biotechnology #sc-2005) or anti-rabbit (SantaCruz Biotechnology #sc-2357) secondary antibodies for 1 h at room temperature. Secondary antibodies were diluted 1:2500 in PBS-Tween 0.1% containing 1% BSA. Proteins were detected with Clarity Western ECL (BioRad #1705061). Amersham Imager 600 luminometer (Amersham, Buckinghamshire, UK) was used for image acquisition. β-actin was used as a control for total protein lysates.

### 2.7. Real-Time PCR

Total RNA was purified from HPFs and HOFs using the RNeasy Plus Mini Kit (Qiagen, Hilden, Germany, #74134), according to manufacturer’s instructions, and quantified at NanoDrop Microvolume Spectrophotometer and Fluorometer (Thermo Fisher Scientific, Waltham, MA, USA). 500 ng of total RNA was used for cDNA synthesis with the QuantiTect Reverse Transcription Kit (Qiagen #205311) and the MJ Mini Personal Thermal Cycler (Bio-Rad), according to manufacturer’s instructions. mRNA expression levels were quantified by Real-Time PCR (RT-PCR) using QuantiFast SYBR Green PCR kit (Qiagen #204054). The nucleotide sequences of the specific primers (Thermo Fisher Scientific) used were: 

IL-6: 5′-TCAAACTGCATAGCCACTTTCC-3′(forward), 5′-AGTTCCTGCAGTCCAGCCTGAG-3′ (reverse); 

IL-8: 5′-CTGGCCGTGGCTCTCTTG-3′(forward), 5′-TTAGCACTCCTTGGCAAAACTG-3′ (reverse); 

VEGF-A: 5′-TACCTCCACCATGCCAAGTG-3′ (forward), 5′-ATGATTCTGCCCTCCTCCTTC-3′ (reverse); 

MMP-3: 5′-TTCCTGGCATCCCGAAGTGG-3′ (forward), 5′-ACAGCCTGGAGAATGTGAGTGG-3′ (reverse); 

β2M: 5′-CCACTGAAAAAGATGAGTATGCCT-3′ (forward), 5′-CCAATCCAAATGCGGCATCTTCA-3′ (reverse); 

qRT-PCR was performed using CFX96TM Touch Real-Time PCR Detection System (Bio-Rad), according to the manufacturer’s instructions. Data were normalized on β2 microglobulin (β2M).

### 2.8. Gelatine Zymography

20 μL of CM from control or DTX/CPT-treated fibroblasts were added to the sample buffer (SDS 0.4%, 2% glycerol, 10 mM Tris-HCl, pH 6.8, 0.001% bromphenol blue) and loaded on a 7.5% SDS-PAGE gel with 0.1% (*w*/*v*) type A-gelatine. Gelatin zymography was then performed as previously described [22].

### 2.9. Total ROS Production

HPFs were incubated with 10 μg/mL of 2′,7′-dichlorodihydrofluorescein diacetate (H2DCF-DA, Invitrogen by Thermo Fisher Scientific #D399) at 37 °C for 30 min. Cells were then lysed in 200 μL RIPA lysis buffer and centrifuged at 14,000 rpm and 4 °C for 10 min. 150 μL of cell lysates were transferred in a 96-well plate and fluorescence was measured at wavelength excitation/emission 485 nm/535 nm using the Biotek Synergy H1 microplate reader. Obtained results were normalized on total protein content quantified with the BCA assay.

### 2.10. Mitochondrial ROS (mtROS) Measurements

HPFs and HOFs were stained with 2.5 µM MitoSOX (Invitrogen by Thermo Fisher Scientific #M36008) in serum-free medium for 15 min at 37 °C, and then cells were trypsinized and resuspended in complete media. After two washing in PBS, cells were collected, and fluorescence was analyzed at wavelength excitation/emission 510 nm/580 nm using BD FACSCanto™ II Flow Cytometry System (BD Bioscience, Franklin Lake, NJ, USA).

### 2.11. Mitochondrial Mass Evaluation

After incubation with 30 nM MitoTracker Green (Invitrogen by Thermo Fisher Scientific #M7514) in serum-free media for 15 min at 37 °C, fibroblasts were trypsinized and resuspended in complete media. Cells were washed in PBS and fluorescence was measured at wavelength excitation/emission 490 nm/516 nm using BD FACSCanto™ II Flow Cytometry System (BD Bioscience).

### 2.12. Mitochondrial Staining in Live Cells

HPFs were seeded in sterile Lab-Tek Chambered Coverglass #1 (Thermo Scientific #155383) and mitochondria were stained with 30 nM MitoTracker Green (Invitrogen by Thermo Fisher Scientific) in serum-free medium for 15 min at 37 °C. Nuclei were labeled by incubating cells with 1 µg/mL Hoechst (Merck-Sigma #H6024) in serum-free media for 5 min. Fluorescence was analyzed using TCS SP8 confocal microscope (Leica, Wetzlar, Germany).

### 2.13. Mitochondrial Membrane Potential Determination

HPFs were incubated with 200 nM TMRE (Invitrogen by Thermo Fisher Scientific #T669) in serum-free medium for 20 min at 37 °C, trypsinized, and resuspended in a complete medium. Cells were washed in PBS and fluorescence was measured at wavelength excitation/emission 549 nm/575 nm using BD FACSCanto™ II Flow Cytometry System (BD Bioscience).

### 2.14. Seahorse XF96 Metabolic Assay

5 × 10^3^ HPFs were seeded in XFe96 cell culture plates (8 technical replicates for each condition) in 80 μL of complete medium and incubated overnight at 37 °C in a humidified atmosphere with 5% CO_2_. 1 h before the analysis, cells were incubated with XF DMEM pH 7.4 supplemented with 1 mM pyruvate, 2 mM glutamine, 10 mM glucose at 37 °C in atmospheric CO_2_ conditions. The Seahorse XF Cell Mito Stress Test (Agilent, Santa Clara, CA, USA #103015-100) was used to measure OCR and EACR parameters, according to the manufacturer’s instructions. The following mitochondrial drugs that modulate cell respiration by targeting the complexes of the electron transport chain were utilized: 1.5 µM oligomycin; 0.5 µM FCCP; 0.5 µM Rotenone/Antimycin A. The drugs were consequently injected three times at the reported time points. Data were normalized on total protein content and quantified with the BCA assay after cell lysis with 10 mM Tris-HCl pH 7.4 and 0.1% Triton X-100.

### 2.15. OCR Measurements with Clarke-O_2_ Type Electrode

Senescent and control fibroblasts were collected by trypsinization and resuspended in a complete medium. After centrifugation at 1000 rpm for 5 min, the cell pellet was resuspended in 1 mL of complete medium and equilibrated at 37 °C. The oxygen consumption rate was measured using a Clarke-type O_2_ electrode (Oxygraph, Hansatech Instruments, Norfolk, UK). 1 mL of cell suspension was added to the chamber, and nmol of O_2_ consumed in a function of time was calculated using O_2_ View Software version 2.09 (Hansatech Instruments). The rate of decrease in oxygen content was normalized on cell number and correlated to cell respiratory capacity.

### 2.16. Lactate Quantification Assay

CM was collected from control or DTX/CPT treated fibroblasts, clarified by centrifugation, and diluted 1:5 in sterile water. Lactate concentration was measured using K-LATE L-Lactic Acid Assay Kit (Megazyme), according to the manufacturer’s instructions. Measurements were normalized on cell number.

### 2.17. Migration and Invasion Assays

Prostate cancer cells (6 × 10^4^ cells for migration and 8 × 10^4^ for invasion) and 1 × 10^5^ ovarian cells resuspended in 200 μL of CM were seeded in the upper compartment of 8 µm-pore-Transwells (6.5 mm diameter) (Greiner Bio-One, Kremsmunster, Austria) pre-coated (invasion assay) or not (migration assay) with 12.5 µg/filter (50 μg/cm^2^) of Matrigel (Corning, Corning, NY, USA). In the lower compartment, 500 μL of DMEM supplemented with 10% FBS was added as a chemoattractant. After 16 h of incubation, the inserts were removed and the non-migrating cells on the upper side of the filter were mechanically removed with a cotton swab. Cells that had migrated toward the lower surface of the filters were then fixed, incubating the inserts for 5 min in ethanol and then stained for 15 min at room temperature with crystal violet 0.5% in 20% methanol. Photographs at randomly chosen fields were taken and cells were counted using ImageJ imaging software (1.53t, Wayne Rasband and contributors, National Institute of Health, USA).

### 2.18. Colony Formation Assay

After 72 h of incubation with CM, 1 × 10^3^ tumor cells were seeded into six-well plates and cultured for 9–11 days. Cells were then fixed and stained with an aqueous solution containing 1% crystal violet and 10% methanol. Colonies were photographed and counted using the ImageJ imaging system.

### 2.19. IHC and Lipofuscin Staining

#### 2.19.1. Prostate Cancer

IHC and lipofuscin staining were performed on human prostate tissue sections. Prostate cancer patients were selected from the Innsbruck Uro-biobank. The use of archived material was approved by the Ethics Committee of the Medical University of Innsbruck. Written consent was obtained from all patients and documented in the database of the University Hospital Innsbruck in agreement with statutory provisions. Two different cohorts of patients were analyzed. The control cohort comprises untreated prostate cancer patients subjected to radical prostatectomy without previous chemotherapy (*n* = 10); the chemotherapy group comprises prostate cancer patients who underwent neo-adjuvant chemotherapy with Taxotere (*n* = 10) before radical prostatectomy. Both patient groups were matched for Gleason Score and age [50].

For each prostate cancer patient, four consecutive tissue sections with 5 μm cutting thickness were obtained for analysis. H&E and p63-α-methylacyl-CoA racemase (AMACR) IHC double staining were performed for the basal characterization of prostate cancer tissue samples. IHC was carried out on a Discovery-XT staining device (Ventana, Tucson, AZ, USA), using the following antibodies: anti-P504S (AMACR) (1:100, Dako, CA, USA) and anti-p63 (4A4) (1:100; Roche, Basel, Switzerland) for prostate samples.

#### 2.19.2. Ovarian Cancer

IHC and lipofuscin staining were performed on high-grade serous ovarian cancer tissues from the archival tissue of Histopathology and Molecular Diagnostics, Careggi Teaching Hospital, Florence. Enrolled patients signed the written informed consent approved by the Tuscany Region Ethics Committee. Two different cohorts of patients were analyzed. The control cohort comprises untreated patients subjected to primary cytoreductive surgery without previous chemotherapy (*n* = 10); the chemotherapy group comprises high-grade serous ovarian cancer patients who underwent chemotherapy with carboplatin AUC 5/6 + paclitaxel 175 mg/m^2^ (*n* = 10). Anti Pax-8 (Cell Marque Corporation, CA, USA #760-4618 Clone: EP 331) staining was performed on ovarian samples.

For lipofuscin staining, tissue slides were incubated for 8 min with 10 mg/mL Sudan Black B (SBB) (Merck Sigma #199664) in ethanol 70%. The nuclear fast red (NFR) (Merck Sigma #60700) counterstaining was performed by incubating the slides with 0.1% NFR for 10 min. Quantification of lipofuscin staining was performed by taking photographs at randomly chosen fields for each slide and measuring black areas in each photo (ImageJ imaging software).

The connective tissue was visualized by Mallory’s trichromic staining performed on distinct tissue sections previously stained with SBB.

### 2.20. Statistical Analyses

Statistical data analysis was performed with GraphPad Prism version 9.0 (GraphPad Software is part of the Dotmatics group, San Diego, CA, USA)). Specifics about statistical tests and post-tests are described in the figure legends. The sample size for all experiments was chosen empirically and independent experiments were pooled and analyzed together. Data are presented as mean ± SEM of at least three independent experiments as indicated in the figure legends. For experiments involving ex vivo cultures of HPFs and HOFs experimental replicates have been performed using fibroblasts from three different patients affected by benign prostatic hyperplasia and four patients with ovarian cancer. *p*-value < 0.05 was considered statistically significant; ns, not significant. * *p* < 0.05, ** *p* < 0.01, *** *p* < 0.001, **** *p* < 0.0001. Mathematical outliers were detected using Grubb’s test (alpha = 0.1) and the values identified were removed.

## 3. Results

### 3.1. Chemotherapy Treatment Induces the Senescent Phenotype in Prostate and Ovarian Cancer Stromal Compartment

To assess the effect of chemotherapy (DTX and CPT) on the stromal compartment of prostate and ovarian cancer, human fibroblasts, isolated from surgical explants of patients affected by benign prostatic hyperplasia (HPFs) or from healthy peritoneal tissues during cytoreductive surgery for ovarian cancer (HOFs), were treated for 24 h with 5 nM DTX and 20 μM CPT respectively, and subsequently incubated in the absence of the drug for the following 6 days. Chemotherapy exposure caused a stable cell growth arrest in both stromal models (Figure 1A,B), and induced the acquisition of a senescent phenotype as indicated by increased β-Gal staining (Figure 1C,F) and DNA-damage foci accumulation (Figure 1G,H), two widely described markers of cellular senescence. In agreement, DTX and CPT-treated cells also displayed a remarkable up-regulation of the senescence marker proteins p21 and p16 Cyclin-Dependent Kinase Inhibitors (Figure 1I–L). Moreover, the qRT-PCR analysis revealed an increase in the expression of major SASP components, such as cytokines IL-6 and IL-8, the growth factor VEGF-A, and the protease MMP-3 in DTX and CPT-treated fibroblasts with respect to untreated controls (Figure 2A,B). In keeping, MMP-2 activity was strongly up-regulated in DTX-treated fibroblasts with respect to control cells (Figure 2C) and a similar trend was observed in CPT-treated HOFs (Figure 2D). Collectively, these data show that chemotherapy exposure induces a senescent phenotype in the stromal compartment of prostate and ovarian cancer, independently of the drug-specific mechanism of action.

### 3.2. Therapy-Induced Senescence (TIS) Rewires Fibroblast Metabolism

Senescent cells frequently display alterations in mitochondrial morphology, dynamic, and functionality [51]. In particular, the senescent phenotype is generally associated with an increased abundance of mitochondria, which however appear to be less functional, resulting in decreased mitochondrial membrane potential, boosted proton leak, and increased generation of reactive oxygen species (ROS) [52,53,54]. Coherently, we observed a striking increase in total ROS content in DTX-treated HPFs (Figure 3A), indicating possible deregulation of the mitochondrial respiratory chain during TIS. In accordance, by specifically quantifying mitochondrial superoxide accumulation, we observed an increase in mitochondrial oxidative stress in DTX-treated HPFs (Figure 3B) and CPT-treated HOFs (Appendix A), confirming an impairment of the respiratory chain following TIS induction. To evaluate possible alterations in mitochondrial mass in senescent fibroblasts, we labeled cells with MitoTracker Green fluorescent dye which localizes inside mitochondria regardless of their membrane potential. Interestingly, an increase in mitochondrial mass in DTX and CPT-treated fibroblasts compared to untreated cells was detected (Figure 3C and Appendix A). To further investigate the mitochondrial functionality in TIS fibroblasts, we quantified the mitochondrial potential of chemotherapy DTX-treated fibroblasts by labeling cells with the cell-permeant, positively charged dye TMRE, which specifically accumulates in active mitochondria. Interestingly, despite the increase in the mitochondrial mass we do not observe an increase in the TMRE staining and hence in mitochondrial functionality (Figure 3D), suggesting that TIS induction correlates with an accumulation of dysfunctional mitochondria in fibroblasts. 

To deeper investigate the mitochondrial activity underpinning TIS induction, we performed a real-time quantification of mitochondrial oxygen flux by performing Seahorse XF96 MitoStress analysis (Appendix A) and Clarke-O_2_ type electrode measurements. Interestingly, while we observed a significant increase in total (Appendix A) and basal (Figure 3E) oxygen consumption rate (OCR) in chemotherapy-treated fibroblasts, we did not see changes in ATP production-linked OCR (Figure 3F), suggesting the inefficiency of mitochondria in senescent cells. Accordingly, by measuring OCR in the presence of the ATP-synthase inhibitor oligomycin, we observed increased proton leak in DTX-treated HPFs (Figure 3G), further confirming an imbalance in mitochondrial efficiency following TIS induction. Together, these data indicate that chemotherapy-induced senescence leads to mitochondrial dysfunction in prostate and ovarian fibroblasts, resulting in ROS accumulation and enhanced oxidative stress. Reduced mitochondrial functionality frequently correlates with enhanced glycolytic metabolism as a compensatory mechanism [55]. In accordance, we observed increased basal acidification rate (ECAR) in DTX–treated fibroblasts (Figure 3H) and extracellular lactate accumulation (Figure 3I and Appendix A) in fibroblasts following TIS induction. Collectively, these data suggest that DTX and CPT stromal exposure results in decreased mitochondrial functionality and increased oxidative stress, forcing fibroblasts to undergo metabolic reprogramming toward glycolysis.

### 3.3. TIS-Fibroblasts Support Prostate and Ovarian Cancer Aggressiveness

To evaluate whether senescent fibroblasts may positively affect tumor progression by enhancing aggressive features of neighboring cancer cells, we evaluated the effect of incubating tumor cells with CM from TIS fibroblasts. Interestingly, we observed that incubation with CM from HPFs and HOFs for 72 h can enhance the migratory and invasive abilities of prostate cancer (PC3 and DU-145) (Figure 4A,B and Appendix A) and ovarian cancer (SKOV-3 and A2780) (Figure 4D,E, and Appendix A) cells, respectively. Moreover, CM from TIS fibroblasts also sustain the clonogenic potential of prostate and ovarian cancer cells as evaluated by colony formation assay (Figure 4C,F and Appendix A). Collectively, these data point out a key role of stromal TIS induction in enhancing cancer cell aggressiveness in prostate and ovarian cancer models.

To better clarify the relevance of stromal senescence in promoting tumor cell aggressiveness, we adopted the senolytic drug ABT-263 to selectively eliminate TIS prostate and ovarian fibroblasts. As expected, 72 h of exposure with 1.25 µM ABT-263 promotes the clearance of senescent fibroblasts as shown by β-Gal staining assay (Figure 4G,H and Appendix A). To investigate whether the removal of senescent fibroblasts is sufficient to impair the previously observed TIS-induced acquisition of malignancy features in prostate and ovarian cancer cells, we incubated PC3 and SKOV-3 cells with CM from TIS fibroblasts treated or not with ABT-263. Interestingly, the treatment with ABT-263 resulted in a reduction of the potential of TIS fibroblast-derived CM to increase the motility abilities of PC3 and SKOV-3 cells (Figure 4I and Appendix A). These data suggest that senolytic drugs, by removing senescent cells in the stromal compartment, may interfere with the senescence-induced malignant phenotype of cancer cells, resulting in the attenuated aggressive potential of treated tumors. Collectively, these results demonstrate that stromal TIS strongly supports the acquisition of aggressive traits by cancer cells and the selective removal of senescent fibroblasts is sufficient to revert these pro-malignant effects.

### 3.4. Chemotherapy Induces Senescence in Tumor Tissues from Post-Therapy Patients

Finally, to assess the clinical relevance of the identified role of stromal TIS in cancer progression, we analyzed a set of paraffin-embedded tissue sections from prostate and ovarian cancer patients treated or not with neo-adjuvant Taxotere or Carboplatin chemotherapy, respectively, before radical prostatectomy or ovarian debulking. By evaluating the presence of senescent cells by lipofuscin staining in paraffin-embedded tissue sections, we observed a strong increase in senescence-positive spots in the stromal, non-tumor tissue compartment of samples from prostate patients treated with chemotherapy prior to radical prostatectomy compared to prostate cancer patients undergoing radical prostatectomy without neo-adjuvant chemotherapy (Figure 5A,B). Mallory’s trichrome and SBB staining confirm that lipofuscin staining is specifically detectable in the stromal compartment of Taxotere-treated patients (Figure 5C). Similarly, lipofuscin accumulation is detected in the stromal/Pax8 negative compartment of Carboplatin-treated ovarian cancer patients compared to untreated ones (Figure 5D,E). Collectively, this evidence demonstrates that stromal TIS has a pro-tumorigenic effect on cancer cells in different tumor types. Strikingly, as shown, this phenomenon can be observed in clinical settings and hence may interfere with the behavior of neighboring cancer cells.

## 4. Discussion

Tumor relapse, treatment failure, and severe and long-term adverse effects still remain the major obstacles in cancer patients’ therapy and emerging evidence is highlighting the key role of cellular senescence in this process [56]. Accelerated aging and an increase in senescence biomarkers, associated with long-term risk of developing chronic illnesses and secondary tumors, have been described in childhood cancer survivors who were exposed to cytotoxic therapies [57]. Accordingly, clinical studies of breast cancer survivors who received chemotherapy or radiotherapy revealed an upregulation of several senescence markers after treatment [58,59].

Besides the widely accepted role of cancer cell oncogenic mutations in tumor progression, the TME is a key player in the complex process of tumorigenesis and metastasis by establishing bidirectional and dynamic crosstalk with tumor cells [60,61]. In this scenario, the importance of the TME in regulating the response to standard-of-care therapy is now widely appreciated [62,63]. Indeed, systemic anticancer treatments not only exert direct cytotoxic effects on tumor cells, but can also indirectly damage the healthy microenvironment, thereby impairing therapy sensitivity and promoting bystander diseases or even cancer relapse [64,65]. Noteworthy, several anticancer interventions act in both cancer cells and TME components as inducers of the senescent phenotype, the so-called TIS [24,66].

In this context, data obtained in the present study are shedding light on a new role of TIS in the stromal compartment of prostate and ovarian tumors in promoting cancer progression and dissemination. In particular, we observed that exposure to chemotherapy treatments induces senescence in prostate and ovarian primary fibroblasts, resulting in the activation of a secretory phenotype and in the rewiring of cellular metabolism. Interestingly, the acquisition of a senescent phenotype strengthens the pro-tumorigenic and pro-metastatic potential of both prostate and ovarian fibroblasts.

Therefore, we propose that the systemic administration of an anti-neoplastic treatment may paradoxically sustain the success of cancer cells that are not fully eradicated by chemotherapy through the establishment of a pro-tumorigenic senescent microenvironment. In agreement with our data, it has been demonstrated that GRO-1-induced senescent fibroblasts promote the progression of ovarian epithelial cancer cells [67]. In addition, in vitro, human prostate fibroblasts, secrete paracrine factors able to sustain adjacent prostate epithelial hyper-proliferation and promote an immunosuppressive TME, mainly through amphiregulin secretion [68,69]. Moreover, it has been shown that the ability of ex vivo primary fibroblasts in promoting the growth of pre-neoplastic prostate epithelial cells positively correlates with the age of the donors [70].

While past research has mainly highlighted the effects of DTX and CPT on tumor cells [40,71], the effects that such treatments may have on the stromal compartment still remain to be clarified. Among the numerous cellular components of the TME, CAFs are crucial actors during all the steps of tumorigenesis, thanks to the secretion of bioactive molecules which regulate tumor occurrence, development, metastasis, and therapeutic resistance [72]. Interestingly, senescent fibroblasts share many similarities with CAFs, and both strongly support the development of a pro-tumorigenic environment [22,73,74,75,76,77]. Indeed, it has been demonstrated that senescent CAFs are present in tumors and may stimulate cancer progression [78,79]. In this light, we can expect that in vivo chemotherapy treatment may affect both the healthy stroma and exacerbate the pro-tumoral features of pre-existing CAFs.

Many studies describe the ability of different antineoplastic agents in inducing the senescence program in tumor cells. However, little is known about the effects of DTX and CPT administration in non-cancerous cells, namely the microenvironmental fibroblasts.

In this study, we investigated the TIS-inducing ability of different anti-neoplastic treatments (DTX and CPT) in the stromal compartment from diverse tumor models (prostate and ovarian cancer cells). Despite their different mechanisms of action, these treatments demonstrated a similar ability to induce senescence in the surrounding stroma, which in turn exerted similar pro-aggressive behavior in the corresponding tumor cells. These data are in line with results shown by several studies indicating that different antineoplastic treatments, in addition to chemotherapy (i.e., kinases and CDK inhibitors, monoclonal antibodies, and radiotherapy) can induce TIS [24]. This widespread phenomenon actually makes senescence a very attractive target for the clinic supporting the value of the recently proposed “one-two-punch strategy”. This therapeutic approach consists of the consequent administration of traditional chemotherapies and senotherapeutics, that selectively revert the negative side effects of anticancer drugs [80]. Interestingly, several studies investigated the effects of different classes of senolytics using pre-clinical models, proving their efficacy for the treatment of many types of cancer but also for the wide spectrum of aging-related diseases [10,46]. Therefore, clinical trials are currently ongoing to study senescent cells as therapeutic targets of senolytic drugs. In this context, a first-in-human pilot study demonstrated the efficacy of an intermittent combination of two senolytics, Dasatinib and Quercetin, in treating iodopathic pulmonary fibrosis, associated with the accumulation of senescent cells [81], providing evidence for the use of senolytics in the clinic. This combination is also being tested in chronic kidney disease (NCT02848131), and in hematopoietic stem cell transplant survivors (NCT02652052). As far as the one-two punch strategy is concerned, Navitoclax treatment in combination with various senescence-inducing chemotherapies entered Phase I and Phase I/II clinical trials. Navitoclax effects have been tested in combination with cisplatin and etoposide in small cell lung cancer (NCT00878449), gemcitabine (NCT00887757), paclitaxel (NCT00891605), docetaxel (NCT00888108), erlotinib (NCT01009073) and sorafenib (NCT02143401) in patients with solid cancers, dabrafinib and trametinib in BRAF mutant metastatic melanoma patients (NCT01989585), and osimertinib in EGFR positive non-small cell lung cancer patients (NCT02520778). Moreover, Navitoclax entered a Phase II clinical trial in combination with the senescence inducer rituximab. This study demonstrated that the combined therapy has higher effectiveness for patients with chronic lymphocytic leukemia (NCT01087151). Despite these promising results, major side effects on hematological cells were reported for Navitoclax [82] and objective responses were often not recorded. Another senolytic, acting as an mTOR inhibitor, Temsirolimus, has been tested in combination with rituximab in a Phase II clinical trial in diffuse large B cell lymphoma [83] (NCT01653067) and in combination with capecitabine chemotherapy in Phase I clinical trial in advanced solid tumors [84] (NCT01050985). These exploratory studies offered a basis to continue the investigation of the “one-two-punch” strategy approach in translational medicine.

Remarkably, in our models, TIS fibroblast treatment with the senolytic drug ABT-263 is sufficient to impair the pro-tumoral effects of senescent stromal cells. The data herein presented opens the possibility of exploiting this approach in the treatment of prostate and ovarian cancers in view of reducing tumor relapse and improving the therapeutic options for these types of tumors.

In keeping, by inquiring about clinical samples from prostate and ovarian patients subjected to Taxotere and Carboplatin-based therapies respectively, we confirmed the senescence-inducing effects of these chemotherapeutics in the stromal compartment.

Collectively, these data point out that the side effects observed following DTX and CPT treatment in prostate and ovarian cancer patients, respectively, may be promoted, at least in part, by the induction of a senescent phenotype in the stromal compartment, generating a supportive TME that further sustains cancer progression and aggressiveness. In this scenario, the selective clearance of DTX- and CPT-induced senescent stromal cells by administering senolytic compounds, could represent a novel promising strategy to improve chemotherapy efficacy and reduce cancer recurrence following chemotherapy.

## Figures and Tables

**Figure 1 cells-11-04026-f001:**
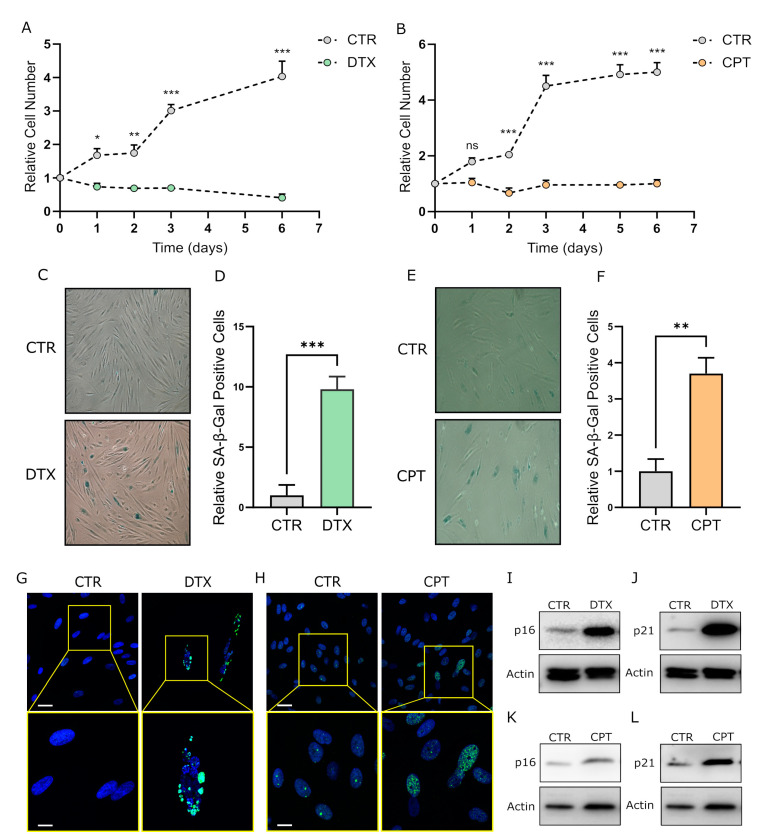
Treatment with DTX or CPT induces the senescent phenotype in primary human prostate and ovarian fibroblasts. (**A**,**B**) Proliferation curves of HPFs (**A**) and HOFs (**B**) treated with 5 nM DTX and 20 μM CPT respectively, and then incubated in culture media without the drugs for further 6 days. Cell number was assessed every 24 h starting from the addition of the chemotherapeutics. Two-way ANOVA with Sidak correction (*n* ≥ 3). (**C**,**E**) SA-β-Gal staining of HPFs and HOFs treated as reported in A-B. SA-β-Gal positive cells are stained in blue. Images are representative of at least three independent experiments. (**D**,**F**) Quantification of SA-β-Gal staining in DTX-treated HPFs (**D**) and CPT-treated HOFs (**F**). For each condition, photos at ten randomly chosen fields were taken and total and blue (SA-β-Gal positive) cells were counted. Data are reported as the relative mean ratio between positive and total cells. Student’s *t* test (*n* = 3). (**G**,**H**) γ-H2AX foci accumulation in DTX-treated HPFs and CPT-treated HOFs. Immunofluorescence on control or chemotherapy-treated fibroblasts was performed using the anti-H2AX phosphorylated on Ser139 (γ-H2AX) antibody (green) and representative confocal microscope images are reported. DAPI was used to stain nuclei (blue), scale bar: 5 μm (**upper panels**); scale bar: 2 μm (**lower panels**). (**I**–**L**) p16 and p21 protein levels in DTX-treated HPFs ((**I**,**J**) respectively) and CPT-treated HOFs ((**K**,**L**) respectively). β-actin was used as a loading control. Representative images of three independent experiments are reported. Data are represented as mean ± SEM of n independent experiments. ns, not significant, * *p* < 0.05; ** *p* < 0.01; *** *p* < 0.001.

**Figure 2 cells-11-04026-f002:**
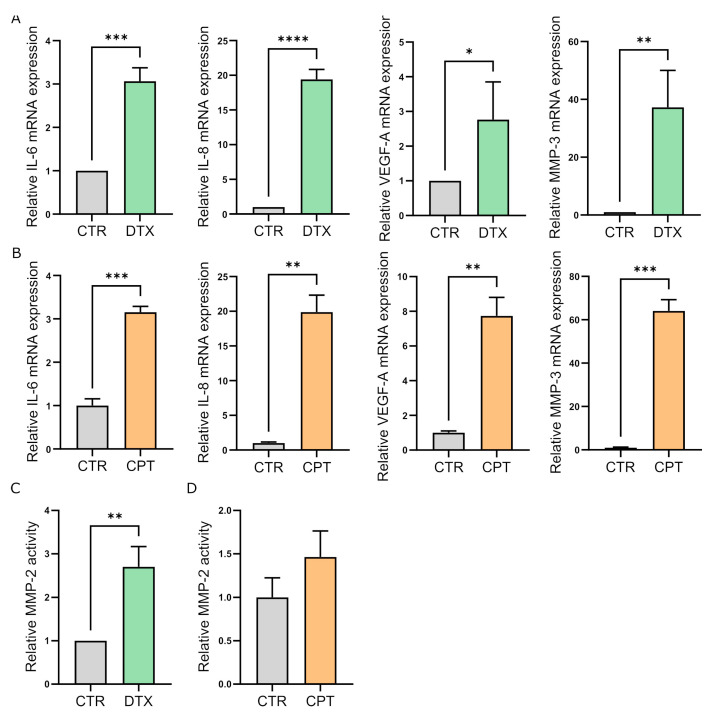
DTX and CPT treatment promotes the development of the senescence-associated secretory phenotype in prostate and ovarian fibroblasts. (**A**,**B**) IL-6, IL-8, VEGF-A, and MMP-3 mRNA levels in DTX-treated HPFs (**A**) and CPT-treated HOFs (**B**). mRNA expression levels were analyzed by quantitative RT-PCR. Student’s *t* test (*n* = 3). (**C**,**D**) MMP-2 activity in fibroblasts upon exposure to DTX (**C**) or CPT (**D**) was evaluated by gelatin zymography. Densitometry analysis of areas of gelatinase activity was performed using the ImageJ imaging system. Student’s *t* test (*n* = 3). Data are represented as mean ± SEM of n independent experiments. * *p* < 0.05; ** *p* < 0.01; *** *p* < 0.001; **** *p* < 0.0001.

**Figure 3 cells-11-04026-f003:**
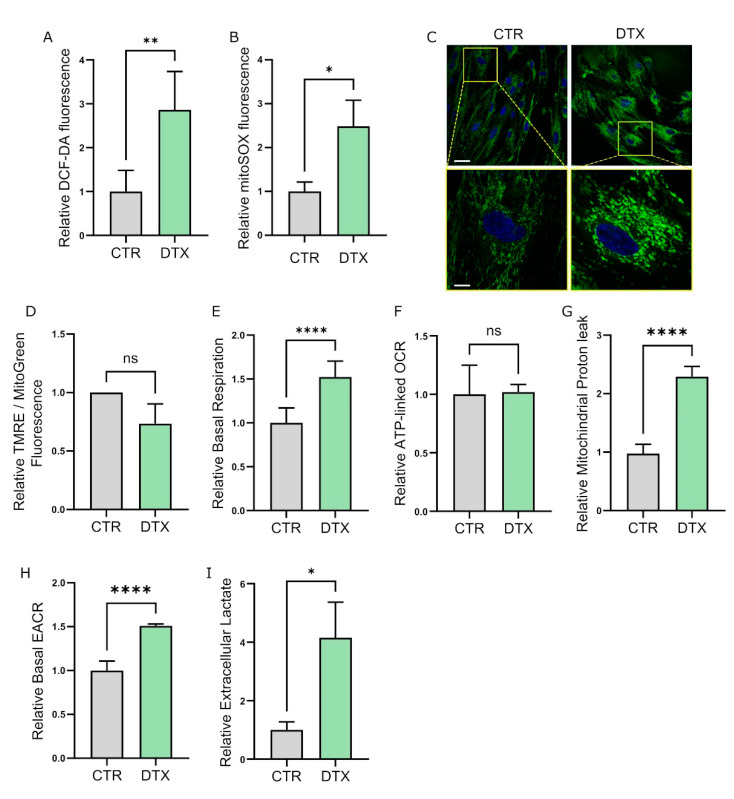
Chemotherapy induces mitochondrial dysfunction and metabolic reprogramming in TIS-fibroblasts. (**A**) Total ROS production in DTX-treated HPFs. Total intracellular ROS production was evaluated by incubating cells with an H2DCF-DA probe and then performing fluorimetric analysis. Data were normalized on total protein content. Student’s *t* test (*n* = 4). (**B**) Mitochondrial ROS production in DTX-treated fibroblasts. Fibroblasts were stained with MitoSOX and fluorescence was evaluated by FACS analysis. Student’s *t* test (*n* = 4). (**C**) Mitochondrial mass evaluation in control or TIS-HPFs. Fibroblasts were stained with MitoTracker Green and representative confocal microscope images of mitochondria are shown. Nuclei were labeled with Hoechst (blue), scale bar: 5 μm (upper panels); scale bar: 2 μm (lower panels). (**D**) Mitochondrial membrane potential of control or DTX-treated HPFs. Cells were incubated with TMRE and the mitochondrial membrane potential was measured by FACS analysis. Data were normalized on MitoTracker Green Fluorescence measured by FACS analysis. Student’s *t* test (*n* = 3). (**E**–**H**) Oxygen consumption rate (OCR) in control and DTX-treated HPFs. Seahorse Mitostress Test analysis was performed in real-time on live control or TIS-fibroblasts. Basal respiration (**E**), ATP-linked respiration (**F**), mitochondrial proton leak (**G**), and basal EACR (**H**) were obtained by measuring OCR following the administration of oligomycin, FCCP, and a mixture of rotenone and antimycin-A. Data were normalized on protein content. Student’s *t* test (*n* ≥ 3). (**I**) Extracellular lactate levels quantification in control or DTX-treated HPFs. CM was collected from control or TIS-fibroblasts and clarified by centrifugation. Results were normalized on cell number. Student’s *t* test (*n* = 3). Data are reported as mean ± SEM of n independent experiments. ns, not significant * *p* < 0.05; ** *p* < 0.01; **** *p* < 0.0001.

**Figure 4 cells-11-04026-f004:**
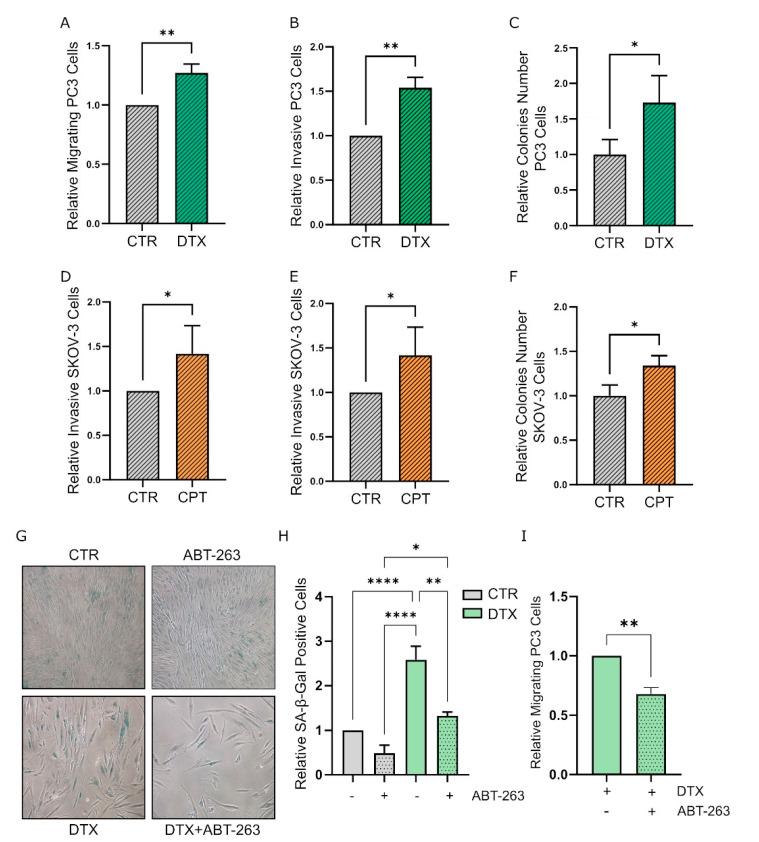
TIS fibroblasts sustain the acquisition of aggressive features in prostate and ovarian cancer cells. (**A**,**B**,**D**,**E**) Migratory (**A**,**D**) and invasive (**B**,**E**) abilities of PC3 (**A**,**B**) and SKOV-3 (**D**,**E**) cancer cells treated with CM from TIS fibroblasts. Tumor cells were incubated for 72 h with CM from control or senescent fibroblasts, before being seeded in the upper compartment of 8 μm Transwell systems pre-coated (**B**,**E**) or not (**A**,**D**) with Matrigel. Cells were let to migrate/invade for 16 h. Migrated/invaded cells were stained with crystal violet and counted (five randomly chosen fields). Student’s *t* test (*n* ≥ 3). (**C**,**F**) Colony formation potential of prostate and ovarian cancer cells. PC3 (**C**) or SKOV-3 (**F**) cancer cells were incubated with CM from control and senescent fibroblasts for 72 h before seeding them in new dishes and incubating for 10 days. Colonies were stained with crystal violet dye, photographed, and counted. Student’s *t* test (*n* = 3). (**G**) SA-β-Gal staining of HPFs, following exposure to chemotherapy and the senolytic drug ABT-263. Senescent cells exposed to the senolytic ABT-263 detach from the plates and are washed away with PBS before staining. Representative images of control, DTX-treated, ABT263-treated, or DTX + ABT263-treated prostate fibroblasts incubated with SA-β-Gal staining solution are reported. (**H**) Quantification of SA-β-Gal staining in control, DTX-treated, ABT263-treated, or DTX + ABT263-treated HPFs. For each condition, photos at ten randomly chosen fields were taken and total and blue (SA-β-Gal positive) cells were counted. Data are reported as the relative mean ratio between positive and total cells. One-way ANOVA with Tukey correction (*n* = 3). (**I**) Migratory abilities of PC3 cancer cells incubated with CM from fibroblasts treated with DTX or DTX + ABT263. After 72 h of incubation with CM from fibroblasts, PC3 cells were seeded in the upper compartment of 8 μm-Transwell and let to migrate toward the lower compartment for 16 h. Cells were stained with crystal violet and counted (five randomly chosen fields). Student’s *t* test (*n* = 3). Data are reported as mean ± SEM of n independent experiments. * *p* < 0.05; ** *p* < 0.01; **** *p* < 0.001.

**Figure 5 cells-11-04026-f005:**
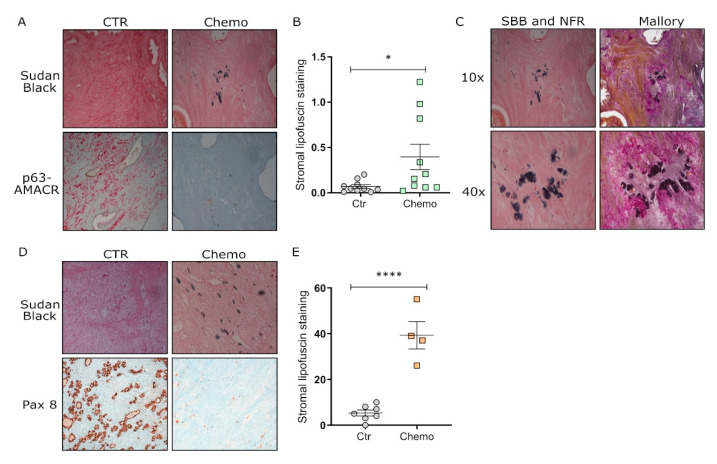
Lipofuscin accumulation is increased in the stromal compartment of chemotherapy-treated prostate and ovarian cancer patients. (**A**) Representative light microscopy images (10×) of double staining with nuclear fast red (NFR)/Sudan Black B (SBB) for the detection of lipofuscin accumulation and p63/AMACR double staining for the identification of epithelial basal prostate cells and prostate cancer cells in control (**left panel**) and chemotherapy group (**right panel**); (**B**) Quantification of stromal lipofuscin staining in all the tissue sections from control (*n* = 10) and chemotherapy (*n* = 10) patient groups was performed using ImageJ imaging software (Unpaired-T test across the control and chemotherapy patient groups, * *p* < 0.05). (**C**) representative light microscopy images (10× top; 40× bottom) of double staining with nuclear fast red (NFR) and Sudan Black B (SBB) for the detection of lipofuscin accumulation in a tissue section from the chemotherapy group (**left panel**) and Mallory’s trichrome and SBB staining for the specific identification of the stromal compartment (**right panel**). Images were taken on two consecutive tissue slides from the same patient. (**D**) Representative light microscopy images (10×) of double staining with nuclear fast red (NFR)/Sudan Black B (SBB) for the detection of lipofuscin accumulation and Pax 8 staining for the identification of epithelial neoplastic cells in control (left panel) and chemotherapy group (right panel). (**E**) Quantification of lipofuscin staining in all the tissue sections from control (*n* = 4) and chemotherapy (*n* = 7) patient groups was performed using ImageJ imaging software (Unpaired-T test across the control and chemotherapy patient groups, **** *p* < 0.0001).

## Data Availability

The data presented in this study are available in the article and in the Appendix A Section. Data are also available on request from the corresponding author.

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
