# Peer review of "Therapy-Induced Stromal Senescence Promoting Aggressiveness of Prostate and Ovarian Cancer"

_cells, 2022, doi:10.3390/cells11244026_

Round 1

Reviewer 1 Report

In this work, Pardella et al. describe how therapy-induced senescence of stromal fibroblasts can enhance the aggressiveness of prostate and ovarian cancer cells.

Generally, the paper is well written, the project is well conceived and executed.

Nevertheless, there are some issues.

MAJOR ISSUES

- Chapter 2.1: since tumor-derived fibroblasts are a key point of this work, the protocol used to derive them from prostate and ovarian cancer specimens should be clearly outlined. Saying that "HPFs and HOFs were isolated from surgical explants" does not enable experiments' reproducibility.

Moreover, an important point when working with patient-derived samples is to clearly define how many patient-derived coltures are used. Authors state, in chapter 2.20, that "at least three independent experiments" were performed for each assay. Anyway, they do not specify if these triplicates were performed with fibroblasts derived from the same patient but at different moments or if they were performed with fibroblasts derived from different patients. The second option could be more appreciable since it could reflect interpersonal variability, but at the same time, clinical and histopathological features of the used samples should be carefully reported to ensure that the cohort is homogeneous and appropriate.

- Fig 1G, this DNA-damage evaluation should be performed also in HOFs

- In chapter 3.2, the vast majority of the experiments presented in this section are performed only on HPFs, this weakens the overall relevance of these findings. Repeating some of these assays also in HOFs would strengthen Authors' theory.

- In lines 444-447, the treatment schedule is unclear. Probably, fibroblasts were previously treated with DTX or CPT and then with ABT-263, but this was not clarified. Also, Authors state that TIS was "removed" by ABT-263, this means that TIS are killed by this treatment? so they detach from plates and are washed away before B-gal staining? Please, specify these details for easier understanding

- In line 571, Authors mention the "one-two-punch strategy" again. This is an interesting point which may be explored further to strengthen the relevance of Authors' findings. Is this strategy already used in some clinical trials? Are these kind of drugs already used in clinical practice or did they have passed phase 1/2 clinical trials?

MINOR ISSUES

- lines 340-342: up-regulation of MMP2 activity is significant only for HPFs and, for both samples, gel images are blurred and unclear. Authors should consider removing the evaluation of the MMP2 activity or, at least, the gel images.

- In chapter 3.2, Authors should provide the original graphs from Seahorse assays. If not as a replacement for the histograms proposed, at least as supplementary figures to ease readers' evaluation of the differences between CTRs and treatments

- Fig. 4C and Suppl.Fig. 2C,F, these images are unclear, blurred and brightness and contrast have been adjusted too high. Authors should evaluate to remove these images

- Suppl.Fig.3, Authors should move it to the main Figure

Reviewer 2 Report

The tumor microenvironment (TME) consists not only of tumor cells and cancer stem cells, but also many other cells, including fibroblasts, stellate cells, immune cells, ECM and collagen, and even microorganisms. All anticancer treatments should affect all cells in the TME and play an important role in cancer pathogenesis and progression. Treatment-induced senescence, TIS, has been recognized as an important issue in cancer progression. However, most research has focused on this effect on cancer cells rather than non-cancer cells. In this study, the authors show that docetaxel or cisplatin induces cellular senescence and promotes invasiveness in human prostate and ovarian fibroblasts, respectively. These results are impressive and lead me to believe that chemotherapy drugs force tumor-associated fibroblasts to senesce. However, it still has several comments on the article:

It is well known that the tumor microenvironment is very complex, and an in vitro single-cell type culture system is beneficial for analyzing signal transduction, but not for more complex situations. Organoid systems (at least studies with co-culture systems of tumor cells and fibroblasts) and animal models will improve these observations. The authors could compare the differences in senescence levels before and after chemotherapy.

2. The authors show the results of chemotherapy-induced tumor tissue senescence in patients in Figure 5, where the chemotherapy strategy was a combination of platinum and paclitaxel. As the authors mention in the Discussion section, they are still obtaining clinical samples from patients treated with DTX and CPT, respectively, and I think observations from these future samples are more appropriate and convincing to present in this paper. In addition, no significant stromal lipofuscin staining was observed in at least six samples from chemotherapy patients in Figure 5B. This also shows the complexity of TME.

However, the novelty and importance of this article is still worthy of publication.

Round 2

Reviewer 1 Report

The authors correctly addressed all the issues.

I would recommend the paper for publication.

Author Response

We thank this reviewer for the acceptance of the manuscript, we are now resubmitting the latest version of the manuscript according to the minor revisions suggested by the Academic Editor. As requested, we added in the Discussion section (marked in red) two  comments about the development of next steps of the study in  more complex and appropriate models and about the importance of increasing the number of samples for lipofuscin staining.
We hope that the present version of the manuscript will be suitable for publicatin in "Cells"
Best Regards
M.Letizia Taddei